# Vitamin C Mitigates Oxidative Stress and Behavioral Impairments Induced by Deltamethrin and Lead Toxicity in Zebrafish

**DOI:** 10.3390/ijms222312714

**Published:** 2021-11-24

**Authors:** Emanuela Paduraru, Elena-Iuliana Flocea, Carlo C. Lazado, Ira-Adeline Simionov, Mircea Nicoara, Alin Ciobica, Caterina Faggio, Roxana Jijie

**Affiliations:** 1Doctoral School of Geosciences, Faculty of Geography-Geology, “Alexandru Ioan Cuza” University of Iasi, Bd. Carol I, 700505 Iasi, Romania; emanuelapaduraru19@yahoo.com (E.P.); mirmag@uaic.ro (M.N.); 2Department of Biology, Faculty of Biology, “Alexandru Ioan Cuza” University of Iasi, Bd. Carol I, 700505 Iasi, Romania; iuliana.flocea@yahoo.com (E.-I.F.); alin.ciobica@uaic.ro (A.C.); 3Nofima, Norwegian Institute of Food, Fisheries and Aquaculture Research, 1433 Ås, Norway; carlo.lazado@nofima.no; 4Multidisciplinary Research Platform (ReForm)-MoRAS Research Center, “Dunarea de Jos” University Galati, 800008 Galati, Romania; ira.simionov@gmail.com; 5Department of Chemical, Biological, Pharmaceutical and Environmental Sciences, University of Messina, Viale F. Stagno d’Alcontre 31, 98166 Messina, Italy; 6Department of Exact and Natural Sciences, Institute of Interdisciplinary Research, “Alexandru Ioan Cuza” University of Iasi, Bd. Carol I, 700505 Iasi, Romania

**Keywords:** deltamethrin, lead, vitamin C, protective role, zebrafish, 3D locomotion analysis

## Abstract

Environmental contamination from toxic metals and pesticides is an issue of great concern due to their harmful effects to human health and the ecosystems. In this framework, we assessed the adverse effects when aquatic organisms are exposed to toxicants such as deltamethrin (DM) and lead (Pb), alone or in combination, using zebrafish as a model. Moreover, we likewise evaluated the possible protective effect of vitamin C (VC) supplementation against the combined acute toxic effects of the two toxicants. Juvenile zebrafish were exposed to DM (2 μg L^−1^) and Pb (60 μg L^−1^) alone and in combination with VC (100 μg L^−1^) and responses were assessed by quantifying acetylcholinesterase (AChE) activity, lipid peroxidation (MDA), some antioxidant enzyme activities (SOD and GPx), three-dimension locomotion responses and changes of elements concentrations in the zebrafish body. Our results show that VC has mitigative effects against behavioral and biochemical alterations induced by a mixture of contaminants, demonstrating that it can be used as an effective antioxidant. Moreover, the observations in the study demonstrate zebrafish as a promising in vivo model for assessing the neuroprotective actions of bioactive compounds.

## 1. Introduction

In recent years, heavy metals and organic compound contamination in aquatic and terrestrial ecosystems has drawn increasing attention due to the detrimental effects on living organisms [1,2,3,4,5]. Environmental surveys have shown the co-occurrence of toxic metals and pyrethroids in water, sediments and fish samples [6,7,8,9]. Lead (Pb) is a toxic metal whose widespread use has resulted in extensive environmental contamination and health problems around the world. The World Health Organization lists Pb in its top ten chemicals of major public health concerns [10]. Although the use of Pb has been significantly reduced in fuel, paint, plumbing and food container solder, resulting in a substantial decline in population exposure to this toxic metal, significant sources of exposure and remanent Pb still exist, especially in developing and poor countries. For example, Pb was found in the bottom sediments of water bodies in the upper Silesia region (southern Poland) in amounts ranging from 32 to 3340 ppm [11]. In water samples from e-waste recycling sites in Guiyu of China, Pb concentration even reached as high as 400 μg L^−1^ [12]. An independent testing at six Washington DC public schools in 2008 showed that 2–41% of sampled tap water has a Pb contamination problem, with the highest lead detected at 1987 μg L^−1^, significantly higher than the EPA-recommended maximum level of 20 μg L^−1^ for schools [13]. As it is described in literature, in addition to its major neurotoxicity [14,15,16,17,18,19] the exposure to Pb has been linked to oxidative stress [20], cardiovascular toxicity [21], as well as gut microbiota dysbiosis and hepatic metabolic disorder [12] in zebrafish. Among the Pb-induced zebrafish neurobehavioral changes identified to date, learning and memory deficits, a decrease in the exploratory activities, altered social and aggressive behavior, color preference or responses to mechanosensory and visual stimuli are included [22,23,24,25,26]. It is very interesting to note that males showed a greater number of attacks on the mirror image versus females at higher exposure regimens (1 and 10 μM Pb), while females displayed a greater number of attacks versus males only at the lower treatment level (0.1 μM Pb) [22]. According to Xu and colleagues, chronic exposure to Pb induced learning impairments that persisted for at least three generations [27]. On the other hand, no significant changes were found in social interactions in adult zebrafish exposed to 50 μg L^−1^ PbCl_2_ [25]. Pyrethroids are synthetic insecticides derived from the natural pyrethrins, that are used for controlling various insect pests in agriculture, and are also used as veterinary drugs and domestic insecticides [28]. Due to their low toxicity and short half-life, these insecticides are being used as viable substitutes for other biocides, such as organophosphates and organochlorines. Pyrethroids are divided in two groups, type I (i.e., alletrin, permethrin, piretrin) and type II (deltamethrin, spermethrin) α cyano group. Deltamethrin (DM) is one of the most popular type II synthetic pyrethroids because of its photostability and high activity against a broad spectrum of insect pests [29,30]. As a result of their widespread use and high lipophilicity, DM can easily reach the aquatic ecosystems and cause serious environmental and health problems. For example, DM is found in environmental samples, such as water and sediments, food, organisms and even human samples (e.g., blood and urine) [31,32,33,34,35,36,37]. The DM concentration in water samples ranged from ng L^−1^ to μg L^−1^. Previous studies have reported that DM exposure could induce zebrafish developmental delay and malformation [38,39,40], apoptosis [38,40], cardiovascular toxicity [38], gonadotoxicity [41] as well as changes in neurochemical, behavioral and cognitive endpoints [20,38,39,42,43,44]. Hyperactivity and swimming towards the surfaces were observed in adult zebrafish after DM (>0.1 μg L^−1^) exposure [39,45]. An increase in the thigmotaxis of zebrafish larvae (preference for the walls) has been observed after treatment with low concentrations of DM (0.1, 1 and 10 μg L^−1^), while at high concentrations (25 and 50 μg L^−1^) the activity decreased [38]. A similar result was found by Hu et al. [43] where they documented significantly reduced average swimming speed of zebrafish larvae exposed to DM at 25 μg L^−1^. Moreover, the results showed that the locomotor activity of zebrafish larvae and the transcript levels of the components of estrogenic and dopaminergic pathways were altered by DM exposure. This was corroborated by Kung et al., where DM exposure during the embryonic period (3–72 hpf) causes hyperactivity in larval zebrafish, which is likely mediated by dopaminergic dysfunction [42].

Although multiple contaminants at low or higher levels coexist and may interact in the environment, most of the published results are focusing on the effects of single chemicals. The evaluation of single pollutant acute and chronic effects does not offer a close to reality view of their impact against aqueous ecosystems. The few studies that evaluated the combined toxicity using zebrafish revealed both synergistic and antagonistic interaction between chemicals [46,47,48,49,50,51]. For example, Our group research previously demonstrated that the toxicity of DM when combined with Cd and Ni significantly decreased as shown by some of the behavioral variables and oxidative stress responses [46]. A similar tendency was reported by Tilton et al. [47], when the addition of copper (Cu) to the moderate dose of chlorpyrifos (CPF) partially attenuated the number of fish undergoing freeze responses. Synergistic responses on zebrafish were observed for binary mixtures of iprodione in combination with pyrimethanil or acetamiprid and ternary mixtures of iprodione + pyraclostrobin in combination with pyrimethanil or acetamiprid [51]. Ku et al. found that the combination of NiSO_4_ and buprofezin formed a complex that facilitated the uptake of compounds by the zebrafish embryos, inducing synergistic effects [50].

Today, zebrafish has become a preferred model organism for ecotoxicological research due to its small size, prolific spawning, rapid life cycle, easy husbandry and low cost, as well because they share many molecular, biochemical, cellular and physiological characteristics with higher vertebrates [52,53,54,55,56]. Additionally, because they exhibit complex behaviors zebrafish may be use in modeling neurobehavioral disorders, to probe the neuroprotective actions of bioactive compounds and to assess the short- or long-term consequences of toxic chemicals exposure [57,58,59]. Although the use of behavioral tests for studying the toxicological effects is relatively a new approach, they have gained significant attention compared with conventional classical methods assessing the growth, lethality and reproductive effects [60,61]. They have attracted interest due to their sensitivity and rapidity, where changes in fish behavioral responses following chemicals exposure may be observed immediately and at relatively low concentrations. The 3D locomotion tracking offers a marked advantage in the existing 2D video tracking, enabling a comprehensive analysis of fish locomotion, interpretation of complex fish behaviors (e.g., anxiogenic vs. anxiolytic manipulations) and detection of subtle behavior alterations [62].

Nowadays there is an increased interest in evaluating the efficiency of dietary products (e.g., vitamin C, vitamin E, quercetin, etc.) in alleviation of pollutants toxicity [63,64,65]. Recent studies have shown that vitamin C (VC, ascorbic acid) co-administration or supplementation in diets of aquatic organisms may reduce the toxicity induced by various pollutants, such as toxic metals, pesticides or pharmaceutical products, acting as a recuperative/protective agent (Table 1). The pretreatment with VC appeared to be a viable option to counter the anxiogenic-like effect and biochemical alterations induced by methylmercury in zebrafish [66]. Similarly, Robea et al. [67] showed that the pretreatment with VC (25 μg L^−1^) significantly attenuated the neurotoxicity of fipronil (Fip, 600 μg L^−1^) and pyriproxyfen (Pyr, 600 μg L^−1^) mixture, as well the oxidative stress. Results revealed that embryonic treatment of zebrafish with VC during embryogenesis enhanced the cell proliferation leading to increased somatic growth in the larval stages, which persisted until the juvenile stage [68]. Butachlor-induced toxicity [69] and neomycin-induced ototoxicity [70] have been ameliorated by VC co-administration, as indicated by decreased mortality, malformation occurrence and the reversion in the response of biomarkers of oxidative stress. VC supplementation was shown to alleviate oxidative stress in tilapia (Oreochromis niloticus) liver and brain tissues induced by chlorpyrifos (CPF) exposure [71]. Only a few species, including humans, guinea pigs, capybara and some birds and fish (i.e.,zebrafish) cannot synthesize VC and this has to be obtained from diet.

In the present study, the risk of exposure to lead and deltamethrin at environmental concentration, as well as the ameliorative effect of vitamin C in juvenile zebrafish were assessed by behavioral and biochemical analyses. To the best of our knowledge, no study has earlier reported the protective role of vitamin C against a heavy metal/pyrethroid insecticide mixture induced toxicity in a zebrafish model. In addition, so far there is no report concerning the application of 3D locomotor activity test for studying the efficiency of VC in alleviation of toxicity induced by a mixture of chemicals.

## 2. Results and Discussion

In order to assess the toxicity of DM and Pb mixtures, as well the protective effects of VC against impairments induced by the pyrethroid insecticide and heavy metal mixture acute exposure against zebrafish juvenile, we performed the three-dimensional (3D) locomotor activity assay. No deaths were found in the control and experimental groups after acute exposure to chemicals. Although variations existed between individuals, distinct spatial differences among the five groups can easily be observed from the 3D swim path reconstructions. As shown in Figure 1, the zebrafish exposed to DM alone displayed an increased top exploration along the perimeter of the tank that turned to normal after 48 h, while the specimens exposed to DM and Pb mixtures exhibited the tendency to swim near the bottom of the tank and gradually started to move to the upper zone after 24 h. Thereby, the time spent in the upper zone indicates an anxiolytic-like behavior and the tendency of zebrafish to swim at the bottom of the tank shows a typical geotaxis, as a result of a high anxiety level [62,76]. Similarly, Egan et al. [77] observed that chronic fluoxetine (100 μg L^−1^) and 0.3% ethanol acute treatments increased the time spent in the top area and the number of transitions to the upper portion of zebrafish. The acute ibogaine (10 mg L^−1^) exposure of zebrafish evoked a short initial top exploration followed by a gradual swimming toward the bottom of the tank [78]. The treatment of adult zebrafish with ZnCl_2_ (0.5–1.5 mg L^−1^) [79] and PbCl_2_ (50 μg L^−1^) [25] for different time periods resulted in reduction in exploratory activity, average speed, entries in the upper zone and elevated freezing duration compared to the control group.

Generally, the time spent as moving/freezing represents the anxiety level. As can be seen in Figure 2A,B, the control and VC groups did not show changes in the total distance traveled during the exposure period, while a significant difference was observed for the total distance swam in the groups exposed to DM alone and in combination with Pb (Figure 2C,D). In addition, both groups presented a significant increasing trend in distance swam and velocity during acute exposure period, as a physiological adaptation of zebrafish to chemicals. The lower value of total distance traveled was measured 6 h after exposure for pyrthroid and heavy metal mixture-treated group. No behavioral change was depicted 48 h after exposure to DM alone. The exposure to DM without and with Pb increased the immobile mean time from an average of 10% of the time for control and VC groups to an average of 60% and 64% of the time, respectively. Based on the zebrafish behavioral endpoints and 3D swim path reconstruction, the DM associated with Pb elicited a slightly more pronounced effects.

Considering the multifaceted role played by vitamin C, we investigated the effectiveness of ascorbic acid in reducing the impairments caused by DM and Pb mixture acute exposure against zebrafish juvenile. As illustrated in Figure 2E, the VC supplementation significantly ameliorated the DM + Pb− induced alteration in zebrafish 3 D locomotion. Similarly to the control and VC-treated groups, the zebrafish covered the entire tank in their exploration. In addition, simultaneous acute exposure to DM *+* Pb mixture and vitamin C has normalized the values of kinematic parameters, within initial limits. For example, the co-administration of VC increased the moving duration measured at 6 h after exposure of zebrafish juvenile with 41%.

As a result of exposure to environmental pollutants, the fine equilibrium between reactive oxygen species (ROS) production and removal by the antioxidant defense systems may be impaired and oxidative damage may occur in the organism [80]. It is known that fish have an efficient antioxidant defense system, composed of various antioxidant enzymes (e.g., superoxide dismutase (SOD), glutathione peroxidase (GPx), glutathione S-transferase (GST) or catalase (CAT)), as well as other low molecular weight scavengers (e.g., non-protein thiols (NPSH)) [81]. With the aim being to evaluate the potential oxidative stress induced by DM and Pb and the efficiency of VC supplementation to mitigate their harmful effects against zebrafish, the superoxide dismutase (SOD) and glutathione peroxidase (GPx) activities, as well MDA levels were determined. While SOD and GPx constitute the first defense line against the toxic effect of oxygen-derived free radicals, lipid peroxidation is one of the main manifestation of oxidative damage induced by chemical compounds [67]. Interestingly, we found that SOD activity, GPx activity and lipid peroxidation were significantly increased in response to the DM and DM + Pb treatments when compared with the control and vitamin C groups, while the raised enzymes activities and MDA level measured for pyrethroid and heavy metal group were significantly decreased by co-administration of vitamin C (Figure 3B–D). Though the present data could not conclusively deduce whether the ameliorative effect of VC against oxidative stress is through ROS elimination or inhibition of free radical formation, the changes observed indicate that it plays a significant role to the response towards the chemicals. The findings are consistent with previous results, Korkmaz et al. [74] and Datta and Kaviraj [73] evaluating the efficiency of VC supplementation in order to remove harmful effects of pesticides (cypermethrin and deltamethrin) in tissue of *Oreochromis niloticus* and *Clarias gariepinus*.

These following studies documented that the repair was not complete and a prolonged recovery period was necessary to return to the basal state. Moreover, it has been found that VC may regulate the production of ROS through Wnt10b signaled in the gill of zebrafish [72]. Wu and colleagues studied the protective role of VC on neomycin-induced ototoxicity in zebrafish and demonstrated that the uptake of neomycin was not influenced by ascorbic acid and they assumed that the inhibition of excessive ROS production contributed to VC protective activity [70].

Acetylcholinesterase (AChE), a key enzyme in cholinergic transmission in the nervous system, is an important biomarker in determining the effects of environmental pollutants in fish. Significant decrease in AChE activity in zebrafish brains exposed to DM with and without Pb were observed relative to control group (Figure 3A), also noted previously by other studies [25,40,77,82]. Parlak reported that AChE activity dramatically decreased in a dose-dependent manner as compared with the control group after exposure of zebafish embryos to detamethrin (2.5–50 μg L^−1^) [40]. Thi et al. shown a reduction in AChE activity in fish after exposure to PbCl_2_ (50 μg L^−1^) [25]. In addition, the neurobehavioral impairment observed in the PbCl_2_-exposed zebrafish has been shown to not be primarily associated with the cholinergic system. A strong, as well as a poor relationship between the alteration of AChE activity and behavioral impairments upon short- or long-term exposure to toxic chemicals have been reported in the literature [47,83,84]. Similar to those observed previously, the effects of chemicals mixture on AChE activity were reversed by vitamin C supplementation, also providing a possible explanation for the preservation of swimming behavior.

Although in the literature it is indicated that pyrethroids interfere with the functioning of sodium channel [85] and the heavy metals uptake is mediated by Ca^2+^ transport pathways [86], in our study the acute exposure to DM and Pb alone and in combination with VC have no visible effect on the total sodium or calcium from the zebrafish body. As shown in Table 2, no significant change was measured in element concentrations between the different experimental conditions, possibly due to the short exposure duration. This finding agrees well with our previous results: when the chronic effects of DM and the acute effects of Cd, Ni and DM on zebrafish is studied, the concentration of essential elements involved in biological structure and biochemical processes did not change in zebrafish body mass [44,46].

Finally, our results not only advance the understanding of deltamethrin and lead combined toxicity, but also reveal the efficiency of vitamin C co-supplementation in the alleviation of behavioral and biochemical alterations induced by a mixture of pollutants (Figure 4). Although we proved that VC has mitigative effects, there is a shortcoming concerning the quantification of the chemicals in water and fish tissues. In addition, further research is required into molecular mechanisms to establish a link between the molecular events and behavioral effects, as well on the capacity of zebrafish to recover after exposure to contaminants (post-exposure period).

## 3. Materials and Methods

### 3.1. Ethical Note

The animals were strictly maintained and treated according to EU Commission Recommendation (2007) on guidelines for the accommodation and care of animals used for experimental and other scientific purposes, Directive 2010/63/EU of the European Parliament and of the Council of 22 September 2010 on the protection of animals used for scientific purposes. This experiment has been approved by Ethics Commission from Faculty of Biology, Alexandru Ioan Cuza” University of Iasi, with the registration number 2923/05.10.2021.

### 3.2. Chemicals

Lead standard solution Certipur^®^ (Pb, 1000 mg L^−1^, 119,776), nitric acid 65% Suprapur^®^ (HNO_3_, 100,441), hydrogen peroxide 30% Perhydro^®^ (H_2_O_2_, 107,210), malonaldehyde bis(diethyl acetal) (T9889), 2-thiobarbituric acid (TBA, T5500), trichloroacetic acid (TCA, 91,230), ethanol EMSURE^®^ (159,010), phosphate buffered salin (P4417), tris hydrochloride solution (T2819), acetylthiocholine iodide (ATCh, A5751), 5, 5-dithio-2, 2-nitrobenzoic acid (DTNB, 322123), Bradford Reagent (B6916), Bovine Serum Albumin (A8022), Superoxide Dismutase Determination Kit (SOD, 19160-1KT-F), Glutathione Peroxidase Cellular Activity Assay Kit (GPx, CGP1-1KT) were purchased from Merck, Darmstadt, Germany. The deltamethrin used in the present study is the active compound of a well-known insecticide (DM, 100 g L^−1^) purchased from a local market with quality certification. The vitamin C (VC, ascorbic acid) liquid form was purchased from a local pharmacy. To avoid any conflict of interest for these products, the brand name of the producer was kept under anonymity. We chose to use them in the study in order to investigate a scenario similar to a real case.

The experimental setup was designed according to OECD Guidelines for the Testing of Chemicals. Section 2: Effects on Biotic Systems. Fish acute toxicity test No. 203. After the acclimatization for 2 weeks, wild-type zebrafish (Dario rerio Hamilton, 1822) (*n* = 15 per group) were randomly divided into five groups as following: Group I was the control group, while Group II (fish exposed to VC at a concentration of 100 μg L^−1^), Group III (fish exposed to DM at a concentration of 2 μg L^−1^), Group IV (fish exposed to 2 μg DM L^−1^ and 60 μg Pb L^−1^) and Group V (fish exposed to 2 μg DM L^−1^ and 60 μg Pb L^−1^ in combination with 100 μg VC L^−1^) were the exposed groups housed in 10 L glass tanks with the respective treatment (Figure 5). The tanks had been filled with dechlorinated tap water that was constantly aerated by an air pump. Fish were not fed 24 h prior to and during exposure and the solution was renewed daily to maintain a similar concentration. In addition, before starting the behavioral tests, during 96 h zebrafish were gently hand-netted from their experimental tank to a temporary tank in order to recover from handling stress, and thereafter, transferred to a novel tank for 3D observations. After the testing was completed, fish from the control and the experimental groups were humanely euthanized with cold water between 2 and 4 °C, according to animal welfare regulations [87].

### 3.3. 3D locomotor Activity Test

The behavioral measurements were carried out in a 10 L glass trapezoidal tank filled with 6 L of dechlorinated tap water by using the Track3D module of EthoVisionXT 14 video tracking software (Noldus Information Technology, TheNetherlands, Holland). The experimental setup has been described in detail in our previous paper [46]. After the experimental accommodation for 96 h, each experimental group is studied using the 3D approach over a 4-min period to set the baseline behavior, presented in our study as initial behavior. Then the 3D locomotion tracking was acquired at 6 h, 12 h, 24 and 48 h for each individual and each experimental condition (100 μg VC L^−1^, 2 μg DM L^−1^ without and with 60 μg Pb L^−1^ and their mixture) to demonstrate the acute changes caused by chemicals exposure (Figure 5). The discussions on behavioral response results were for each group with comparison between initial behavior and treatments based on the averages resulting after experimental repetition. In order to assess the pollutants’ effects as well as the protective effect of VC against behavioral impairments, we analyzed the following behavioral endpoints: the total distance moved in the 3D space (m), velocity (mm s^−1^) and freezing duration (s), defined as the absence of the body movements.

### 3.4. Determination of AChE Activity

Following completion of exposure, whole brains were extracted and homogenized on ice in 60 vol. (*v*/*w*) of 0.05 M Tris-HCl, pH 8.0, according to a previously reported protocol [88]. AChE activity in the homogenates was determined according to protocol described by Pan et al. [82]. Briefly, 50 μL sample and 50 μL ATCh (5 mM) were incubated at 30 °C for 15 min to a final volume of 0.1 mL and then the reaction was stopped by 0.125 mM DTNB-phosphate-ethanol reagent inside 0.9 mL (12.4 mg of DTNB dissolved in 125 mL 95% ethanol, 75 mL distilled water and 50 mL 0.1 M PBS, pH 7.5) as the thiol indicator. The color was detected immediately at 412 nm by using the Specord 210 Plus from Analytik Jena, Germany. The results were normalized to the total protein content determined with the Bradford method [89] in which bovine serum albumin was used as standard. The AChE activity (% of control) is used to analyze the effects induced by chemicals. To ensure the consistence of our results, the tests were performed in triplicates.

### 3.5. Determination of Whole Body MDA Concentration, SOD and GPx Activities

After 48 h exposure period, each fish was homogenized in 10 volumes of ice-cold 0.1 M phosphate buffer (pH 7.4) and then centrifuged at 5500 rpm for 15 min (4 °C). The supernatant fraction was used to determine the SOD and GPx activities, MDA level and the protein concentrations in samples. Lipid peroxidation (LPO) was assessed using thiobarbituric acid reactive substances determination method. Briefly, whole-body homogenates (200 μL) were mixed to trichloroacetic acid (0.92 M in 0.25 N HCl) and thiobarbituric acid (26 mM) and vortexed. Afterwards, a 20 min incubation at 100 °C (boiling water bath) and a 10 min centrifugation (3000 rpm) were performed. The supernatants were read at 532 nm and the samples absorbances were compared with a *melon*-dialdehyde (MDA, 0.5–20 nmol mL^−1^) standard curve. Raw data were corrected by the protein content and normalized as percentage of control values. [90] Activities of SOD and GPx were determined colorimetrically using kits (Merck, Darmstadt, Germany), following the manufacturer’s instructions.

### 3.6. Whole Body Ions and Trace Metals Analysis

For the mineralization process, three fish per experimental group were washed several times with ultrapure water, ground, weighed (~0.3 g) and microwave digested (Berghof speedwave MWS-2) with 4 mL of HNO_3_ and 2 mL of 30% H_2_O_2_. Subsequently, the samples were transferred to 50 mL tubes and diluted with ultrapure water until a 50 mL volume was reached. Copper (Cu) concentrations were measured with HR-CS GF-AAS (ContrAA 700-Analytik, Jena, Germany) following the method described by Plavan et al. [91], while an air-acetylene FL-ASS with hollow-cathode lamps (GBC Avanta, Australia) was used to determine zinc (Zn), iron (Fe), calcium (Ca), sodium (Na), magnesium (Mg) and potassium (K) concentrations. Calibration has been accomplished with certified standard solutions (1000 mg L^−1^, from Merck, Darmstadt, Germany). The methods’ accuracy were validated by using a reference material for fish muscle (ERM-BB422), certified by the EU Joint Research Center Institute for Reference Materials and Measurements. From the reference material, 6 samples were prepared using the same digestion protocol. All concentrations were reported as µg g^−1^ wet weight ± SD [46].

### 3.7. Statistical Analysis

Firstly, the normality and distribution of the data were assessed using Graph Pad Prism software (v 9.0, San Diego, CA, USA), calculating Shapiro-Wilk test. Afterwards, comparisons among treatments were performed using one-way ANOVA followed by Turkey’s or Dunnett’s multiple comparisons tests, as indicated in the figure legends. All experiments were replicated at least three times to ensure the reproducibility of the results, the results being expressed as the mean ± standard deviation (SD). The *p*-values were shortened with the following symbols: * *p* < 0.05, ** *p* < 0.01, *** *p* < 0.001 and **** *p* <0.0001 (significance was set at *p* < 0.05). Plots were generated in Graph Pad Prism v9.0. The video trials resulted from the behavioral tests were sorted in files for each group. These were reused and combined in EthoVision XT 14 to generate average heat maps for experiment with replicas. With the Track3D module were extracted data where the average was calculated and represented in 3D figures (the 3D spatio-temporal reconstruction of the swim paths).

## 4. Conclusions

The results of behavioral and biochemical assays reveal that acute exposure to deltamethrin without or with lead may elevate the oxidative stress and anxiety level characterized by increased freezing duration and reduced exploratory activity, as well as may cause a significant decrease in the activity of AChE in zebrafish brain. Particular movement patterns have been observed for zebrafish exposed to DM and DM + Pb. The animals exposed to DM displayed significantly more transitions to and time in the upper half, over the 4 min testing period, while the specimens exposed to mixture spent more time at the tank bottom. Zebrafish exposed to DM and Pb gradually started to move to the upper zone after 24 h. It has been observed that VC co-administration can mitigate the behavioral impairments and the negative effect of oxidative stress caused by chemicals exposure in zebrafish. Treatment with VC reduced the elevated lipid peroxidation, as well the increased levels of SOD and GPx, and restored partially the reduced AChE activity. Together, these results indicate that zebrafish is a good model for testing chemicals present in the aquatic environment and similarly, in investigating compounds, such as vitamin C, that can mitigate the consequences from toxicant exposure. Future studies are necessary for the detailed characterization of the mechanisms underlying both the joint toxic effects and protective role observed.

## Figures and Tables

**Figure 1 ijms-22-12714-f001:**
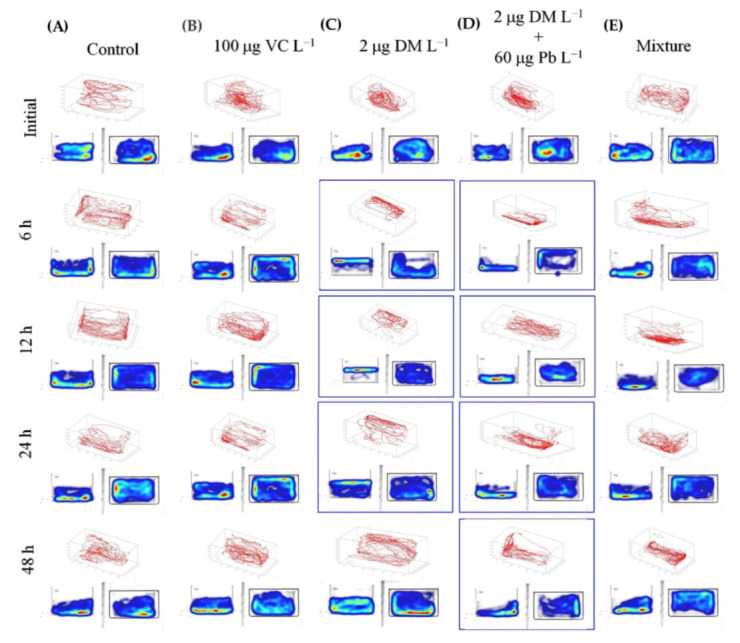
The 3D swim traces results for zebrafish control group (**A**) and groups treated with (**B**) 100 μg VC L^−1^, (**C**) 2 µg DM L^−1^, (**D**) 2 µg DM L^−1^ + 60 µg Pb L^−1^ and (**E**) mixture of VC, Pb and DM at 0, 6, 12, 24 and 48 h (*n =* 15). An automated integration of traces using Track3D software results in 3D swim tracks (red color). The bottom panels show the side and top view of zebrafish in the 4 min test.

**Figure 2 ijms-22-12714-f002:**
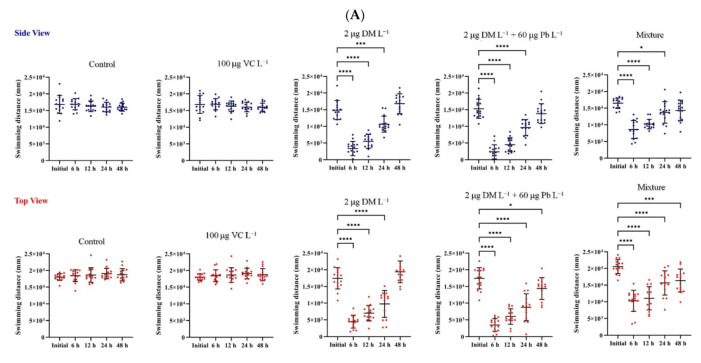
3D locomotor activity assay for control and treated groups with 100 μg VC L^−1^, 2 µg DM L^−1^, 2 µg DM L^−1^ + 60 µg Pb L^−1^ and mixture of VC, Pb and DM. The graphs show (**A**) the total distance in the 3D space covered by the zebrafish (mm), (**B**) the velocity (mm/s) and (**C**) freezing duration (s) under the condition of no exposure and exposure to chemicals. Each dot represents an individual zebrafish (*n* = 15 each experimental group). Data were expressed as mean ± SD. Statistically significant differences are denoted by * *p*< 0.05, ** *p* < 0.01, *** *p* < 0.001, **** *p* < 0.0001 (one-way ANOVA, followed by a Dunnett’s multiple comparisons test).

**Figure 3 ijms-22-12714-f003:**
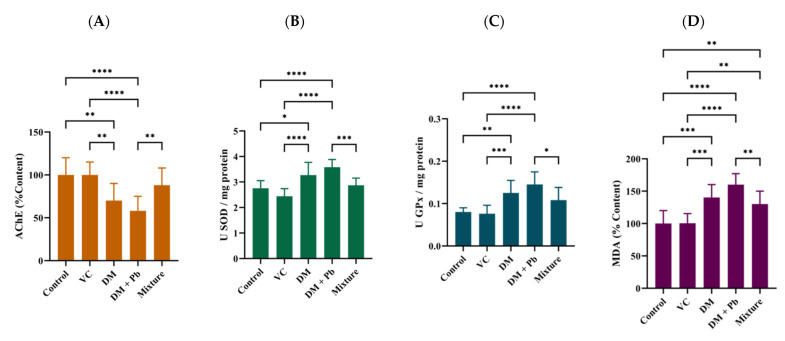
Activity of (**A**) AChE, (**B**) SOD, (**C**) GPx and (**D**) MDA levels in zebrafish following treatment with vitamin C (100 μg L^−1^), DM (2 μg L^−1^) and Pb (60 μg L^−1^) alone and in combination with VC (100 μg L^−1^) for 48 h. Values are expressed as means ± SD (*n* = 9). Statistically significant differences are denoted by * *p* < 0.05, ** *p* < 0.01, *** *p* < 0.001, **** *p* < 0.0001 (one-way ANOVA, followed by a Turkey’s test).

**Figure 4 ijms-22-12714-f004:**
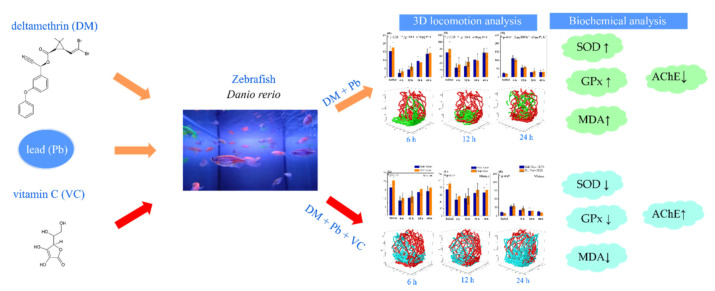
Schematic presentation of the present study. The adverse effects of exposure to environmental pollutants, such as lead and deltamethrin mixture, and the efficiency of vitamin C in alleviation of their toxicity in juvenile zebrafish were assessed. The exposure to a heavy metal/pyrethroid mixture induced behavioral abnormalities and oxidative stress in fish. In addition, the efficiency of vitamin C in alleviation of toxicity caused by chemicals exposure was demonstrated.

**Figure 5 ijms-22-12714-f005:**
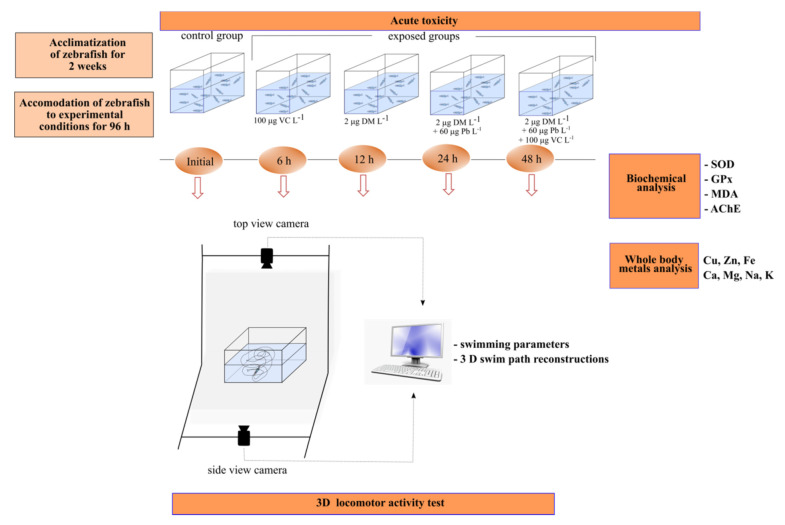
The schematic representation of the experimental setup used to elucidate the potential toxic effect of DM in combination with Pb on zebrafish and its possible amelioration by vitamin C supplementation.

**Table 1 ijms-22-12714-t001:** Summary of fish studies regarding the protective/recuperative role of vitamin C.

	Chemical Exposure	Vitamin C	Exposure Period	Fish	Assessments	Refs.
administration	-	0.5 and 1.0 g kg^−1^ VC diets	6 weeks	zebrafish	- gene expression of Wnt10b - β-catenin - oxidative stress	[72]
pretreatment with VC	1 μg g^−1^ MeHg	2 mg g^−1^ (intraperitoneal)	24 h	adult zebrafish	- light/dark preference - T-4,5-D	[66]
	600 μg L^−1^ Pyr 600 μg L^−1^ Fip	25 μg L^−1^	14 days	zebrafish juvenile	- swimming performance - social behavior - oxidative stress	[67]
dietary supplement of VC	12 and 24 μg L^−1^ CPF	200 mg VC 100 g^−1^ feed	96 h	Nile tilapia, Oreochromis Niloticus	- oxidative stress	[71]
	5 μg L^−1^ DM	100 mg VC 100 g^−1^ feed supplemented diet for 60 days	24 h	catfish, Clarias gariepinus	- hepatosomatic index - liver glycogen - plasma glucose - ascorbic acid level in blood, liver and kidney	[73]
	0.22 μg L^−1^ and 0.44 μg L^−1^ cypermethrin	300 mg VC 100 g^−1^ feed supplemented diet for 20 days	20 days of exposure + 15 days of recuperation	Nile tilapia, Oreochromis niloticus	- histopathological and biochemical methods	[74]
co-administration	200 and 400 μg L^−1^ CPF	200 and 400 μg L^−1^	72 h	zebrafish embryos	- AChE gene transcription and activity - oxidative stress - development	[75]
	1–15 μmol L^−1^ butachlor	40 and 80 μg L^−1^	72 h	zebrafish embryos/larvae	- mortality, malformation rates - oxidative stress	[69]
	20 μM neomycin	0.002–0.02 μg L^−1^ (14.2–113.6 μM)	30 min	zebrafish embryos/larvae	- ototoxicity - oxidative stress	[70]
	2 μg L^−1^ DM and 60 μg L^−1^ Pb	100 μg L^−1^	48 h	juvenile zebrafish	- 3D locomotor activity test - oxidative stress - AchE activity - body elements concentrations	Present study

**Table 2 ijms-22-12714-t002:** Total electrolytes and trace metal profiles reported in zebrafish body measured for wet weight (average ± SD).

Elements (μg g^−1^)	Exposure			
	Control	100 μg VC L^−1^	2 μg DM L^−1^ + 60 μg Pb L^−1^	2 μg DM L^−1^ + 60 μg Pb L^−1^ + 100 μg VC L^−1^
**Cu**	1.23 ± 0.14	1.14 ± 0.04	1.16 ± 0.117	1.22 ± 0.05
**Fe**	20.5 ± 0.6	19.3 ± 1.2	17.5 ± 2.2	18.7 ± 1.5
**Ca**	5586 ± 156	5478 ± 325	5681 ± 225	5521 ± 358
**Mg**	390 ± 46	402 ± 11.6	397 ± 37.6	367 ± 52
**Na**	708 ± 52	644 ± 29	717 ± 59	679 ± 28
**K**	2061 ± 69	1993 ± 57	2062 ± 269	2067 ± 105

The results for the fish muscle reference material that was prepared and analyzed were: Na (certified 2.8 g kg^−1^ and obtained 2.65 ± 1.3 g kg^−1^), K (certified 21.8 g kg^−1^ and obtained 21.1 ± 0.65g kg^−1^), Ca (certified 0.342 g kg^−1^ and obtained 0.343 ± 0.002 g kg^−1^), Mg (certified 1.37 g kg^−1^ and obtained 1.38 ± 0.03 g kg^−1^), Zn (certified 16 ± 1.1 mg kg^−1^ and obtained 15.3 ± 1.2 mg kg^−1^), Fe (certified 9.4 ± 1.4 mg kg^−1^ and obtained 9.3 ± 0.96 mg kg^−1^) and Cu (certified 1.67 ± 0.16 mg kg^−1^ and obtained 1.64 ± 0.32 mg kg^−1^).

## Data Availability

Not applicable.

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
