# Peer review of "Vitamin C Mitigates Oxidative Stress and Behavioral Impairments Induced by Deltamethrin and Lead Toxicity in Zebrafish"

_ijms, 2021, doi:10.3390/ijms222312714_

Round 1

Reviewer 1 Report

In the present manuscript, the adverse effects of exposure to environmental pollutants such as lead and one pyrethroid, deltamethrin, was evaluated. In addition, the ameliorative effect of vitamin C for the combination of these two compounds was assessed. For this the authors used the animal model zebrafish. One major asset of the zebrafish model compare to more traditional cell culture assays, is that it allows analyses of complex behaviors traits used to model neurobehavioral disorders, and to assess the short and long-term consequences of toxic chemicals exposure. Such parameters were nicely explored and analyzed in this study. In parallel, several biochemical parameters were quantified, such as acetylcholinesterase activity, lipid peroxidation, and SOD or GPx antioxidant enzyme activities, showing an increased oxidative status in stressed animals. Finally, it was demonstrated that vitamin C mitigated effects against behavioral and biochemical alterations induced by a mixture of the two chemicals. Together, these results indicate that zebrafish is a good model for testing pollutants present in the aquatic environment and in investigating protective effects of compounds of interests, such as vitamin C.

Generally, this study is well carried out, the biological question is nicely introduced and experiments are carefully conducted. Adequate statistical analyses permitted to extract relevant observations, and conclusions are in agreement with obtained data. However, the following observations might help to improve the general quality of the manuscript.

Introduction part is too long and sounds more like a review than a specific introduction to the subject. It will beneficiate of a more focused presentation of the biological question addressed in the experiments.

It is a shame that lead treatment was not evaluated alone. Therefore it is not possible to conclude whether the combine toxicity of lead and deltamethrin resulted in additive of synergistic effect.

Protein carbonylation would be another easily accessible and interesting marker of increased oxidative stress to be addressed.

Results in table 2 are not presented, neither analyzed. If they are irrelevant the table should be removed. However, unless it is a mistake, the Ca level in DML, Pb and VC treated animals is about ten times the value of others. This requires to be analyzed and comment.

Discussion, the statement “as well as may cause alterations in AChE activity in zebrafish brain, consistent with the cognitive dysfunction observed” is too far interpretation of obtained data. Such conclusion requires dedicated anatomic or physio-pathological evaluation

Minor

Last lane Page 3, “analysess” one extra-S.

Author Response

Reviewer 1

In the present manuscript, the adverse effects of exposure to environmental pollutants such as lead and one pyrethroid, deltamethrin, was evaluated. In addition, the ameliorative effect of vitamin C for the combination of these two compounds was assessed. For this the authors used the animal model zebrafish. One major asset of the zebrafish model compare to more traditional cell culture assays, is that it allows analyses of complex behaviors traits used to model neurobehavioral disorders, and to assess the short and long-term consequences of toxic chemicals exposure. Such parameters were nicely explored and analyzed in this study. In parallel, several biochemical parameters were quantified, such as acetylcholinesterase activity, lipid peroxidation, and SOD or GPx antioxidant enzyme activities, showing an increased oxidative status in stressed animals. Finally, it was demonstrated that vitamin C mitigated effects against behavioral and biochemical alterations induced by a mixture of the two chemicals. Together, these results indicate that zebrafish is a good model for testing pollutants present in the aquatic environment and in investigating protective effects of compounds of interests, such as vitamin C.

Generally, this study is well carried out, the biological question is nicely introduced and experiments are carefully conducted. Adequate statistical analyses permitted to extract relevant observations, and conclusions are in agreement with obtained data. However, the following observations might help to improve the general quality of the manuscript.

Point 1. Introduction part is too long and sounds more like a review than a specific introduction to the subject. It will beneficiate of a more focused presentation of the biological question addressed in the experiments.

Our response: As suggested by the reviewer, we have revised the manuscript for clarity.

Changes in the manuscript:

(Page 2)

As it is described in literature, in addition to its major neurotoxicity [14-19] the exposure to Pb has been linked to oxidative stress [20], cardiovascular toxicity [21], as well as gut microbiota dysbiosis and hepatic metabolic disorder [12] in zebrafish. Among the Pb induced zebrafish neurobehavioral changes identified to date include learning and memory deficits, a decrease in the exploratory activities, altered social and aggressive behavior, color preference or responses to mechanosensory and visual stimuli [22-26]. It is very interesting to note that males showed a greater number of attacks on the mirror image versus females at higher exposure regimens (1 and 10 mM Pb), while female displayed a greater number of attacks versus males only at the lower treatment level (0.1 mM Pb) [22]. According to Xu and colleagues, chronic exposure to Pb induced learning impairments that persisted for at least three generations [27]. On the other hand, no significant changes were found in social interactions in adult zebrafish exposed to 50 mg L-1 PbCl2 [25].

Previous studies have reported that DM exposure could induce zebrafish developmental delay and malformation [38-40] , apoptosis [38, 40], cardiovascular toxicity [38], gonadotoxicity [41] as well as changes of in neurochemical, behavioral and cognitive endpoints [20, 38, 39, 42-44].

(Pages 2-3)

The evaluation of single pollutant acute and chronic effects does not offer a close to reality view of their impact against aqueous ecosystems.The few studies that evaluated the combined toxicity using zebrafish revealed both synergistic and antagonistic interaction between chemicals [46-51]. For example, Our group research previously demonstrated that the toxicity of DM when combined with Cd and Ni, significantly decreased as shown by some of the behavioral variables and oxidative stress responses [46]. A similar tendency was reported by Tilton et al. [47], when the addition of copper (Cu) to the moderate dose of chlorpyrifos (CPF) partially attenuated the number of fish undergoing freeze responses. Synergistic responses on zebrafish were observed for binary mixtures of iprodione in combination with pyrimethanil or acetamiprid and ternary mixtures of iprodione+pyraclostrobin in combination with pyrimethanil or acetamiprid [51]. Ku et al. found that the combination of NiSO4 and buprofezin formed a complex that facilitated the uptake of compounds by the zebrafish embryos, inducing synergistic effects [50].

Today, zebrafish has become a preferred model organism for ecotoxicological research due to its small size, prolific spawning, rapid life cycle, easy husbandry and low cost, as well because they share many molecular, biochemical, cellular and physiological characteristics with higher vertebrates [52-56]. Additionally, because they exhibit complex behaviors zebrafish may be use in modeling neurobehavioral disorders, to probe the neuroprotective actions of bioactive compounds and to assess the short – or long-term consequences of toxic chemicals exposure [57-59]. Although, the use of behavioral tests for studying the toxicological effects is relatively a new approach, they have gained significant attention compared with conventional classical methods assessing the growth, lethality and reproductive effects [60, 61]. They have attracted interest due to their sensitivity and rapidity, where changes in fish behavioral responses following chemicals  exposure may be observed immediately and at relatively low concentrations. The 3D locomotion tracking offers a marked advantage in the existing 2D video tracking, enabling a comprehensive analysis of fish locomotion, interpretation of complex fish behaviors (e.g. anxiogenic vs anxiolytic manipulations) and detection of subtle behavior alterations [62].

Point 2. It is a shame that lead treatment was not evaluated alone. Therefore it is not possible to conclude whether the combine toxicity of lead and deltamethrin resulted in additive of synergistic effect.

 Our response: “Studies on the impact of heavy metals in mixture with other organic compounds on aquatic organisms” is the subject of a new research project.

Point 3. Protein carbonylation would be another easily accessible and interesting marker of increased oxidative stress to be addressed.

 Our response: We agree that the protein carbonylation is another marker which can be used as a measure of oxidative injury and we wish to introduce in our future studies.

Point 4. Results in table 2 are not presented, neither analyzed. If they are irrelevant the table should be removed. However, unless it is a mistake, the Ca level in DML, Pb and VC treated animals is about ten times the value of others. This requires to be analyzed and comment.

 Our response: We apologize for the mistake; we have revised the table and the manuscript.

Changes in the manuscript:

(Pages 11-12)

Although in the literature is indicated that pyrethroids interfere with the functioning of sodium channel [91] and the heavy metals uptake is mediated by Ca2+ transport pathways [92], in our study the acute exposure to DM and Pb alone and in combination with VC have no visible effect on the total sodium or calcium from the zebrafish body. As shown in Table 2, no significant change was measured in element concentrations between the different experimental conditions, possibly due to the short exposure duration. This finding agrees well with our previous results when studied the chronic effects of DM and the acute effects of Cd, Ni and DM on zebrafish the concentration of essential elements involved in biological structure and biochemical processes did not change in zebrafish body mass [44, 46].

Table 2. Total electrolytes and trace metal profiles reported in zebrafish body measured for wet weight (average ± SD).

Elements (mg g-1)

Exposure

Control

100 mg VC L-1

2 mg DM L-1 + 60 mg Pb L-1

2 mg DM L-1 + 60 mg Pb L-1              + 100 mg VC L-1

Cu

1.23 ± 0.14

1.14 ± 0.04

1.16 ± 0.117

1.22 ± 0.05

Fe

20.5 ± 0.6

19.3 ± 1.2

17.5 ± 2.2

18.7 ± 1.5

Ca

5586 ± 156

5478 ± 325

 5681 ± 225

5521 ± 358

Mg

390 ± 46

402 ± 11.6

397 ± 37.6

367 ± 52

Na

708 ± 52

644 ± 29

717 ± 59

679 ± 28

K

2061± 69

1993 ± 57

2062± 269

2067± 105

The results for the fish muscle reference material that was prepared and analyzed were: Na (certified 2.8 g kg-1 and obtained 2.65±1.3 g kg-1), K (certified 21.8 g kg-1 and obtained 21.1±0.65g kg-1), Ca (certified 0.342 g kg-1 and obtained 0.343±0.002 g kg-1), Mg (certified 1.37 g kg-1 and obtained 1.38±0.03 g kg-1), Zn (certified 16±1.1 mg kg-1 and obtained 15.3±1.2 mg kg-1),  Fe (certified 9.4±1.4 mg kg-1 and obtained 9.3±0.96 mg kg-1) and Cu (certified 1.67±0.16 mg kg-1 and obtained 1.64±0.32 mg kg-1).

Point 5. Discussion, the statement “as well as may cause alterations in AChE activity in zebrafish brain, consistent with the cognitive dysfunction observed” is too far interpretation of obtained data. Such conclusion requires dedicated anatomic or physio-pathological evaluation

Our response: We agree with the reviewer and the correction as been made.

Changes in the manuscript:

(Page 11)

Significant decrease in AChE activity in zebrafish brains exposed to DM with and without Pb were observed relative to control group (Figure 4A), also noted previously by other studies [25, 40, 77, 88]. Parlak reported that AChE activity dramatically decreased in a dose-dependent manner as compared with the control group after exposure of zebafish embryos to detamethrin (2.5 – 50 mg L-1) [40]. Thi et al. shown a reduction of AChE activity in fish after exposure to PbCl2 (50 mg L-1) [25]. In addition, the neurobehavioral impairment observed in the PbCl2 exposed zebrafish has been shown to not be primarily associated with the cholinergic system. A strong as well a poor relationship between the alteration of AChE activity and behavioral impairments upon short- or long-term exposure to toxic chemicals have been reported in the literature [88-90].

(Page 13)

The results of behavioral and biochemical assays revealed that acute exposure to deltamethrin without or with lead may elevate the oxidative stress and anxiety level characterized by increased freezing duration and reduced exploratory activity, as well as may cause a significant decrease in the activity of AChE in zebrafish brain.

Minor

Point 6. Last lane Page 3, “analysess” one extra-S.

Our response: The correction has been made.

Reviewer 2 Report

Comments to the Authors

              Paduraru E. and coworkers have revealed that vitamin C plays critical roles, and protective antioxidative effects, against behavioral and biochemical alterations induced by a mixture of deltamethrin and lead toxicity in a promising in vivo model using zebrafish.  The present Figures, Results & Discussion are highly motivated and interesting, from relatively high-quality data in most part. This manuscript seems to match the Journals of molecular science, and this paper likely provides some new insights from those data.

              Before its publication, at the very least, the authors should,

  1. All of the graph bars should be represented with boxes that include individual datapoints (for example, made by GraphPad Prism software).
  2. Revise the Figure 4A together with the denote of statistically significant differences.
  3. What is the most critical point in this paper?? What is new??  It is a little hard to read it especially in the ‘Introduction’ & ‘Results and Discussion’ parts, because those sections are very long and sometimes boring.
  4. Create a diagram (i.e., schematic presentation) depicting the Authors’ critical insights from this study. It will be very helpful to our readers.
  5. Discuss the Authors’ study limitations in detail, upon one paragraph of the Discussion part.

Author Response

Reviewer 2

  Paduraru E. and coworkers have revealed that vitamin C plays critical roles, and protective antioxidative effects, against behavioral and biochemical alterations induced by a mixture of deltamethrin and lead toxicity in a promising in vivo model using zebrafish.  The present Figures, Results & Discussion are highly motivated and interesting, from relatively high-quality data in most part. This manuscript seems to match the Journals of molecular science, and this paper likely provides some new insights from those data.

              Before its publication, at the very least, the authors should,

Point 1. All of the graph bars should be represented with boxes that include individual data points (for example, made by GraphPad Prism software).

Our response: As suggested by the reviewer, we have revised the manuscript and the graph for clarity.

Changes in the manuscript:

(Page 7)

2.4. Statistical analysis

Firstly, the normality and distribution of the data were asessed using Graph Pad Prism v9.0 (GraphPad software, CA, USA), calculating Shapiro-Wilk test. Afterwards, comparisons among treatments were performed using one-way ANOVA followed by Turkey’s or Dunnett’s multiple comparisons tests, as indicated in the figure legends. All experiments were replicated at least three times to ensure the reproducibility of the results, the results being expressed as the mean ± standard deviation (SD). The p-values were shortened with the following symbols: * p<0.05, ** p<0.01, *** p<0.001 and ****p<0.0001 (significance was set at p < 0.05). Plots were generated in Graph Pad Prism v9.0.

(Pages 9-10)

(A)

(B)

(C)

Figure 3. 3D locomotor activity assay for control and treated groups with 100 mg VC L-1, 2 µg DM L-1, 2 µg DM L-1 + 60 µg Pb L-1 and mixture of VC, Pb and DM. The graphs show (A) the total distance in the 3D space covered by the zebrafish (mm), (B) the velocity (mm/s) and (C) freezing duration (s) under the condition of no exposure and exposure to chemicals. Each dot represents an individual zebrafish (n=15 each experimental group). Data were expressed as mean ± SD. Statistically significant differences are denoted by *p<0.05, **p<0.01, ***p<0.001, ****p<0.0001 (one-way ANOVA, followed by a Dunnett’s multiple comparisons test).

Point 2. Revise the Figure 4A together with the denote of statistically significant differences.

Our response: As suggested by the reviewer, we have revised the Figure 4 for clarity.

Changes in the manuscript:

(Page 11)

Figure 4. AChE, SOD, GPx activity and MDA levels in zebrafish following treatment with vitamin C (100 mg L-1), DM (2 mg L-1) and Pb (60 mg L-1) alone and in combination with VC (100 mg L-1) for 48 h. Values are expressed as means ± SD (n=9). Statistically significant differences are denoted by *p<0.05, **p<0.01, ***p<0.001, ****p<0.0001 (one-way ANOVA, followed by a Turkey’s test).

Point 3. What is the most critical point in this paper?? What is new??  It is a little hard to read it especially in the ‘Introduction’ & ‘Results and Discussion’ parts, because those sections are very long and sometimes boring.

Our response: As suggested by the reviewer, we have revised the manuscript for clarity.

Changes in the manuscript:

(Page 2)

As it is described in literature, in addition to its major neurotoxicity [14-19] the exposure to Pb has been linked to oxidative stress [20], cardiovascular toxicity [21], as well as gut microbiota dysbiosis and hepatic metabolic disorder [12] in zebrafish. Among the Pb induced zebrafish neurobehavioral changes identified to date include learning and memory deficits, a decrease in the exploratory activities, altered social and aggressive behavior, color preference or responses to mechanosensory and visual stimuli [22-26]. It is very interesting to note that males showed a greater number of attacks on the mirror image versus females at higher exposure regimens (1 and 10 mM Pb), while female displayed a greater number of attacks versus males only at the lower treatment level (0.1 mM Pb) [22]. According to Xu and colleagues, chronic exposure to Pb induced learning impairments that persisted for at least three generations [27]. On the other hand, no significant changes were found in social interactions in adult zebrafish exposed to 50 mg L-1 PbCl2 [25].

Previous studies have reported that DM exposure could induce zebrafish developmental delay and malformation [38-40] , apoptosis [38, 40], cardiovascular toxicity [38], gonadotoxicity [41] as well as changes of in neurochemical, behavioral and cognitive endpoints [20, 38, 39, 42-44].

(Pages 2-3)

The evaluation of single pollutant acute and chronic effects does not offer a close to reality view of their impact against aqueous ecosystems.The few studies that evaluated the combined toxicity using zebrafish revealed both synergistic and antagonistic interaction between chemicals [46-51]. For example, Our group research previously demonstrated that the toxicity of DM when combined with Cd and Ni, significantly decreased as shown by some of the behavioral variables and oxidative stress responses [46]. A similar tendency was reported by Tilton et al. [47], when the addition of copper (Cu) to the moderate dose of chlorpyrifos (CPF) partially attenuated the number of fish undergoing freeze responses. Synergistic responses on zebrafish were observed for binary mixtures of iprodione in combination with pyrimethanil or acetamiprid and ternary mixtures of iprodione+pyraclostrobin in combination with pyrimethanil or acetamiprid [51]. Ku et al. found that the combination of NiSO4 and buprofezin formed a complex that facilitated the uptake of compounds by the zebrafish embryos, inducing synergistic effects [50].

Today, zebrafish has become a preferred model organism for ecotoxicological research due to its small size, prolific spawning, rapid life cycle, easy husbandry and low cost, as well because they share many molecular, biochemical, cellular and physiological characteristics with higher vertebrates [52-56]. Additionally, because they exhibit complex behaviors zebrafish may be use in modeling neurobehavioral disorders, to probe the neuroprotective actions of bioactive compounds and to assess the short – or long-term consequences of toxic chemicals exposure [57-59]. Although, the use of behavioral tests for studying the toxicological effects is relatively a new approach, they have gained significant attention compared with conventional classical methods assessing the growth, lethality and reproductive effects [60, 61]. They have attracted interest due to their sensitivity and rapidity, where changes in fish behavioral responses following chemicals  exposure may be observed immediately and at relatively low concentrations. The 3D locomotion tracking offers a marked advantage in the existing 2D video tracking, enabling a comprehensive analysis of fish locomotion, interpretation of complex fish behaviors (e.g. anxiogenic vs anxiolytic manipulations) and detection of subtle behavior alterations [62].

In the present study, the risk of exposure to lead and deltamethrin at environmental concentration, as well as the ameliorative effect of vitamin C in juvenile zebrafish were assessed by behavioral and biochemical analyses. To the best of our knowledge, no study has earlier reported the protective role of vitamin C against a heavy metal/pyrethroid insecticide mixture induced toxicity in a zebrafish model. In addition, so far there is no report concerning the application of 3D locomotor activity test for studying the efficiency of VC in alleviaton of toxicity induced by a mixture of chemicals.

Point 4. Create a diagram (i.e., schematic presentation) depicting the Authors’ critical insights from this study. It will be very helpful to our readers.

Our response: As suggested by the reviewer, we have added the requested information.

Changes in the manuscript:

(Page 12)

Figure 5. Schematic presentation of the present study. The adverse effects of exposure to environmental pollutants, such as lead and deltamethrin mixture, and the efficiency of vitamin C in alleviation of their toxicity in juvenile zebrafish were assessed. The exposure to a heavy metal/pyrethroid mixture induced behavioral abnormalities and oxidative stress in fish. In addition, it was demonstrated the efficiency vitamin C in alleviation of toxicity caused by chemicals exposure.

Point 5. Discuss the Authors’ study limitations in detail, upon one paragraph of the Discussion part.

Our response: As suggested by the reviewer, we have added the requested information.

Changes in the manuscript:

(Page 12)

Finally, our results not only advance the understanding of deltamethrin and lead combined toxicity, but also reveal the efficiency of vitamin C co-supplementation in alleviation of behavioral and biochemical alterations induced by a mixture of pollutants (Figure 5). Although, we proved that VC has mitigative effects, there is a shortcoming concerning the quantification of the chemicals in water and fish tissues. In addition, further research is required onto molecular mechanisms to establish a link between the molecular events and behavioral effects, as well on the capacity of zebrafish to recover after exposure to contaminants (post-exposure period).

Round 2

Reviewer 2 Report

This revised version of the Authors' manuscript is improved and sounds very well.  Thanks so much for reading it.